# Technical Note: Design flood under hydrological uncertainty

Anna Botto[1*], Daniele Ganora[1], Pierluigi Claps[1], and Francesco Laio[1]

[1]Department of Environment, Land and Infrastructure Engineering - Politecnico di Torino - Corso Duca degli Abruzzi, 24 - 10129 Torino, Italy
[*]Now at the Department of Civil, Environmental and Architectural engineering - Università di Padova - Via Marzolo, 9 - Padova, Italy

*Correspondence to:* Daniele Ganora (daniele.ganora@polito.it)

**Abstract.** Planning and verification of hydraulic infrastructures demands for a design estimate of hydrologic variables, usually provided by frequency analysis, neglecting hydrologic uncertainty. However, when hydrologic uncertainty is accounted for, the design flood value for a specific return period is no longer a unique value, but is represented by a distribution of values. As a consequence, the design flood is no longer univocally defined, making the design process undetermined.

The Uncertainty Compliant Design Flood Estimation (UNCODE) procedure is a novel approach that, starting from a range of possible design flood estimates obtained in uncertain conditions, converges to a single design value. This is obtained through a cost-benefit criterion with additional constraints that is numerically solved in a simulation framework. This paper contributes to promote a practical use of the UNCODE procedure without resorting to numerical computation. A modified procedure is proposed by using a correction coefficient that modifies the standard (i.e., uncertainty-free) design value on the basis of sample length and return period only. The procedure is robust and parsimonious, as it does not require additional parameters with respect to the traditional uncertainty-free analysis.

Simple equations to compute the correction term are provided for a number of probability distributions commonly used to represent the flood frequency curve. The UNCODE procedure, when coupled with this simple correction factor, provides a robust way to manage the hydrologic uncertainty and to go beyond the use of traditional safety factors. With all the other parameters being equal, an increase of the sample length reduces the correction factor, and thus the construction costs, still keeping the same safety level.

## 1 Introduction

The flood frequency curve is commonly used to derive the design flood as the quantile $Q_T$ corresponding to a fixed return period $T$. For practical reasons, $Q_T$ is commonly expressed only as a single value; however, $Q_T$ can only be expressed in this way if its frequency distribution and its parameters are known perfectly. In practice, one can only estimate the frequency distribution and its parameters using a sample of observed data, thereby inflating the uncertainty in the estimate of $Q_T$. However, the design of an hydraulic infrastructure demands for a single design value to be selected. A gap therefore exists between theory and practice. Quantitative methods to measure the uncertainty associated to the quantiles of the flood frequency curve (e.g., through their variance or probability distribution) have been proposed (e.g., Cameron et al., 2000; De Michele and Rosso,

2001; Brath et al., 2006; Blazkova and Beven, 2009; Laio et al., 2011; Liang et al., 2012; Viglione et al., 2013), but very few suggestions are provided about how to extract a single design value from the probability distribution of possible design values.

Botto et al. (2014), with the development of the Uncertainty Compliant Design Flood Estimation (UNCODE) procedure, have shown that it is possible to select meaningful flood quantiles from their distribution by considering an additional constraint based on a cost-benefit criterion. Hence, the output is a unique design flood value $Q_T^*$. Before illustrating the UNCODE approach, it is worth recalling the working principles of the cost-benefit analysis, which is a core element of the procedure. Cost-benefit analysis can be used to estimate the design flood as the flow value which minimizes the total expected cost function, defined as the sum of the actual cost to build a flood protection infrastructure (cost function) and the expected damages caused by a flood event. An illustrative example of this approach is reported in Fig. 1a. The cost function is rather easy to understand, being an increasing function of the design flood. Instead, the expected damage function needs to be computed point-by-point: for any single tentative design flood value (see the inset in Fig. 1a) it equals the integral of the product of the probability density function (pdf) of the flood flow values and a specific damage function. The latter indicates the damage occurring when the flood exceeds the flow value used to design the infrastructure. The damage function depends on a number of parameters such as the exposure and vulnerability of the flooded goods, the flooding dynamics and the topography, to mention a few. For these reasons the damage function turns out to be very site-specific and often unavailable, due to the lack of information needed to compute it (Menoni et al., 2016); in these cases the cost-benefit method results inapplicable.

To face this problem Botto et al. (2014) made the assumption that costs and damages can be represented by linear functions, with slope $c$ and $d$ respectively, as illustrated in Fig. 1b. Given this assumption, the total cost, $C_{TOT}$, can be computed as

$$C_{TOT} = c \cdot Q^* + \int_{Q^*}^{\infty} d \cdot (Q - Q^*) \cdot p(Q|\mathbf{\Theta}) \, dQ, \tag{1}$$

where $Q^*$ is the generic design flood value and $p(Q|\mathbf{\Theta})$ is the probability density function of the flood flow with parameters $\mathbf{\Theta}$. The optimal design flood of the (uncertainty-free) cost-benefit framework can be then calculated as the value that minimizes Eq. (1). Examples of cost-benefit analysis in the hydrologic/hydraulic context can be found in the literature (Bao et al., 1987; Ganoulis, 2003; Jonkman et al., 2004; Tung, 2005), with only a few of them accounting for uncertainty (Al-Futaisi and Stedinger, 1999; Su and Tung, 2013).

Botto et al. (2014) further demonstrated that the optimal design flood obtained from the cost-benefit analysis with linear cost and damage functions is equivalent to the design flood $Q_T$ obtained from the standard frequency analysis, provided that uncertainty is not accounted for and the ratio between $d$ and $c$ equals the return period $T$. This result can be shown by setting to 0 the derivative of $C_{TOT}$ with respect to $Q^*$, in order to find the minimum of Eq. 1; this leads to the equivalence

$$\frac{d}{c} = \frac{1}{1 - P(Q^*|\mathbf{\Theta})} = T, \tag{2}$$

where $P(\cdot)$ is the cumulative distribution function of the flood values and $T$ is the return period. This is valid provided that the probability distribution used in the cost-benefit framework is the same used in the standard frequency analysis.

The UNCODE approach is founded on the joint use of the cost-benefit approach of Eq. (1) and the constraint derived in Eq. (2). The rationale behind this approach is that it is possible to apply the cost-benefit framework with standard, but meaningful,

cost and damage functions. This is particularly convenient because the cost-benefit framework can be easily extended to include the estimation uncertainty inherent in the limited sample length of hydrological records. Consequently, also the UNCODE framework (which is a particular case of cost-benefit analysis) can be extended to account for this kind of uncertainty. In uncertain condition, the parameters of the flood frequency distribution, $\boldsymbol{\Theta}$, become a random vector; hence, the uncertainty can be included in the cost benefit analysis by compounding $C_{TOT}$ over all the possible values of $\boldsymbol{\Theta}$. In mathematical terms, the cost-benefit framework with uncertainty is summarized by the equation

$$Q_T^* = \underset{Q^*}{\arg\min} \left[ \int_{\boldsymbol{\Theta}} C_{TOT}\left(Q^*|c,d,p(\boldsymbol{\Theta})\right) \cdot h(\boldsymbol{\Theta})\,\mathrm{d}\boldsymbol{\Theta} \right], \tag{3}$$

where $h(\boldsymbol{\Theta})$ is the joint pdf of the parameters of the flood frequency curve. Equation (3) represents the full UNCODE model, which adopts linear cost and damage functions and accounts for uncertainty in a cost-benefit framework.

It is worth noting that, as a consequence of the inherent equivalence of Eq. (2), there are no additional parameters in the cost-benefit framework; in fact, $c$ and $d$ are related through the known value of the return period $T$. The remaining free parameter can be shown to affect only the magnitude of the integral in Equation (3) but not the position of its minimum, thus avoiding the need for further parameters in the UNCODE framework with respect to the standard design flood procedure.

To simplify the UNCODE application, which requires the use of numerical computation of $Q_T^*$, we provide here an approximated, yet reliable method to estimate $Q_T^*$ starting from $Q_T$. Other than a useful practical tool for design purposes, the analysis reported in this note also provides a method to quantify the "value" of newly available hydrological information or the effect of data scarcity on $Q_T^*$ due to uncertainty.

## 2 Practical estimation of the UNCODE design flood

The UNCODE design flood, $Q_T^*$, results in a systematically larger value than its corresponding standard value $Q_T$, as shown by Botto et al. (2014). Moreover, the relative difference between the two values,

$$y = \frac{Q_T^* - Q_T}{Q_T}, \tag{4}$$

has been reported to increase with the return period (as the quantile uncertainty increases) as well as, for fixed $T$, with the standard deviation of the probability distribution of $Q_T$ (i.e., with the uncertainty of $Q_T$). We propose to calculate the approximated estimate of the UNCODE design flood, hereafter referred to as $\hat{Q}_T^*$, directly by inversion of Eq. (4), without resorting to the numerical solution of Eq. (3). This solution reads:

$$\hat{Q}_T^* = (1 + \hat{y}) \cdot Q_T, \tag{5}$$

where the correction factor $\hat{y}$ (i.e., the approximated estimator of $y$) needs to be computed separately. Given this background, we propose to model $\hat{y}$ according to the equation

$$\hat{y} = 10^{-2} \cdot \exp\left[ a_0 + a_1\sqrt{n} + a_2\ln T \right], \tag{6}$$

where $T$ is the return period and $n$ is the sample length which can be considered as a proxy of the standard deviation of $Q_T$; $n$ can be computed from at-site records or as an equivalent sample length from the regional estimate of $Q_T$.

The coefficients $a_0$, $a_1$ and $a_2$ depend on the probability distribution adopted in the frequency analysis. They have been evaluated from an extensive simulation study in which the full UNCODE procedure has been systematically applied to many simulated records, created by combining the following criteria:

1. The parent distribution $\mathcal{P}$ is selected among the most common distributions used in flood frequency: log-Normal (LN3), Generalized Extreme Value (GEV), Generalized Logistic (GLO), Pearson type III (PE3) and log-Pearson type III (LP3). For details on the probability distribution equation and on the relationship between parameters and L-moments the reader is referred to Hosking and Wallis (1997). The LP3 corresponds to the PE3 with log-transformed values.

2. The sample length $n$ of annual maxima is selected from the list: 30, 40, 50, 60, 70 80, 90, 100.

We generated 100 records for each combination of $\mathcal{P}$ and $n$. Looking at the properties of the L-moments, 90% of the synthetic records fall within the ranges: $0.28 \leq$ L-CV $\leq 0.40$, $0.14 \leq$ L-skewness $\leq 0.40$ and $0.07 \leq$ L-kurtosis $\leq 0.32$, which correspond well with values typically encountered in real-world applications. The standard design flood $Q_T$ as well as the (exact) UNCODE estimator $Q_T^*$ have been computed for each record of the simulated dataset. This step has been performed by adopting a suitable fitting distribution $\mathcal{F}$ to the whole synthetic dataset. To make the results more general, $\mathcal{F}$ has been selected from the list: LN3, GEV, GLO, PE3, LP3. Note that any $\mathcal{F}$ is used to fit records from any parent $\mathcal{P}$, as in real cases the exact parent distribution is not known a priori. In this way, the error due to the misspecification of the fitting distribution is included in the results. The correction factor $y$ (Eq. 4) has been computed for all the available records in the simulated dataset and for different return periods $T$ (respectively equal to 50, 100, 200, 500 and 1000 years). It depends on the fitting distribution $\mathcal{F}$ adopted in the frequency analysis. Finally, the exact $y$ values have been regressed against $n$ and $T$ to obtain their estimate $\hat{y}$ (using an ordinary least squares linear regression on the log-transformed terms of Eq. 6). Different forms of Eq. (6) have also been tested, but are not reported as they provide less accurate results.

Coefficients $a_0$, $a_1$ and $a_2$ are reported in Table 1 for different fitting distributions commonly used in the hydrological practice to compute the design flood (in fact, the fitting distribution is always known, while the parent is not). It can be noticed that, when increasing the sample length $n$, the difference between $Q_T^*$ and $Q_T$ is reduced, due to the negative value of the coefficient $a_1$. Table 1 reports also some diagnostics of the regressions used to estimate the coefficients. The global performance of the regressions has been evaluated using the coefficient of determination and residuals analysis (through the mean absolute error, MAE, and root mean squared error, RMSE) for each fitting distribution. The value of the coefficient of determination ranges from 0.96 in case of the PE3 and 0.94 for the LN3, to 0.85 for the GEV and GLO. The MAE and the RMSE take values around 0.02, corresponding to a 2% variation in the design flood estimation, which is negligible in many situations. In general, the PE3 probability distribution results in the best performance in terms of residuals analysis and $R_{adj}^2$ as can be appreciated looking at the results reported in Table 1.

The reliability of the approximated correction factor $\hat{y}$ estimated with the regression model has also been evaluated by comparing the $\hat{Q}_T^*$ value obtained through Eq. (5) and (6) with its exact counterpart calculated with the full UNCODE procedure

(Eq. 3). As a reference, time series listed in Botto et al. (2014, Table 1) with at least 30 years of record length have been analyzed, assuming the LN3 and the GEV as possible fitting distributions and different return periods. Results show a very good agreement between the exact ($Q_T^*$) and the approximated ($\hat{Q}_T^*$) UNCODE design flood values, as reported in Fig. 2, where each panel shows the estimates for all series and all the return periods.

5  A synthesis of the obtained results is shown in Fig. 3, where the values of $\hat{y}$ have been reported for the studied distributions, based on a set of typical sample length and return period values. As mentioned, a direct comparison of the results between different distributions is not possible, but it is relevant to observe that for all the distributions $\hat{y}$ evolves in the same way for varying $n$ and $T$ values. In general, the correction factor does not exceed 10% of the standard value $Q_T$ for intermediate return periods (e.g., $T = 200$ years) even for small samples, although a significative variability is associated to the distribution type.

10 It is around 10% for $T = 500$ years with sample length values ($n = 50$) commonly available at many gauged stations. On the other hand, the sample length plays an important role: for example, considering $T = 500$ years, the GEV distribution and varying the sample size, the reduction of the $y$ value is about 0.075 between $n = 30$ and $n = 50$, and to 0.040 between $n = 50$ and $n = 70$.

## 3 Discussion of the application conditions

15 The UNCODE approach to flood frequency analysis provides a solution to quantify the design flood estimate when considering the uncertainty of the distribution quantile; however, application of the full UNCODE procedure may be cumbersome and computationally demanding for the practitioner. An approximate but reliable framework has been proposed here to allow a easy computation of the UNCODE design flood value from the standard value using a correction factor, $\hat{y}$.

  The extensive simulation analysis at the base of this study shows that the coefficients relating the UNCODE value $\hat{Q}_T^*$ to

20 the traditionally-computed value $Q_T$ are distribution-dependent. For the most used distributions in flood frequency analysis they have been computed and provided. The choice of the distribution and the quantification of its associated uncertainty is a problem of model selection; hence it cannot be solved by the UNCODE procedure, but depends on the methods of standard flood frequency analysis.

  The obtained results demonstrate that an increase in the length of relatively short samples has a noticeable impact in terms of

25 reduction of $\hat{y}$ that results in a reduction of the UNCODE estimate $\hat{Q}_T^*$. This implies that, while the infrastructure keeps the same safety level (or, equivalently, is designed with the same return period), and with all other parameters being equal, additional data reduce uncertainty and consequently the construction costs. The UNCODE design value is indeed reduced with respect to the UNCODE estimate computed with less data. Consequently, the coefficient $\hat{y}$ can be considered a measure of the value of data. The mentioned results agree with findings recently obtained by Ganora and Laio (2016) in a study on the relative role of regional

30 and at-site flood frequency modeling approaches, where the value of at-site data has been highlighted and regarded as a reliable way to improve regional predictions, even with short records. Under this perspective, the correction factor can be used as a metric for uncertainty comparison and quantification, thus providing a further tool to combine different modeling approaches, similarly to the applications of Kjeldsen and Jones (2007) and Ganora et al. (2013) who, with different methodologies, have

exploited measures of hydrologic uncertainty to merge regional and at-site information. Finally, the correction factor is a new and easy-to-implement design tool which provides a quantitative way to determine the design flood value accounting for hydrologic uncertainty, while keeping the same design hazard level considered in standard uncertainty-free analyses. This is a novel approach when compared to the common engineering practice, which accounts for hydrologic uncertainty by considering,

5     for instance, the hydraulic freeboard. The use of the freeboard is equivalent to increasing the design flood value, but without accounting for the size of the system (e.g., the basin area), nor for the hydrologic information available at the section (i.e., observed of equivalent record length used to compute the standard design flood); therefore, this approach is not tailored to the specific case study. The correction factor represents an advance with respect to the use of "all-encompassing" safety factors and towards a clearer way to manage the different sources of uncertainty in hydrological and hydraulic design.

10     *Acknowledgements.* Funding from the ERC Consolidator Grant 2014 n. 647473 "CWASI - Coping with water scarcity in a globalized world" is acknowledged. D. Ganora also acknowledges the RTD Starting Grant from Politecnico di Torino. The work is based on simulated data.

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

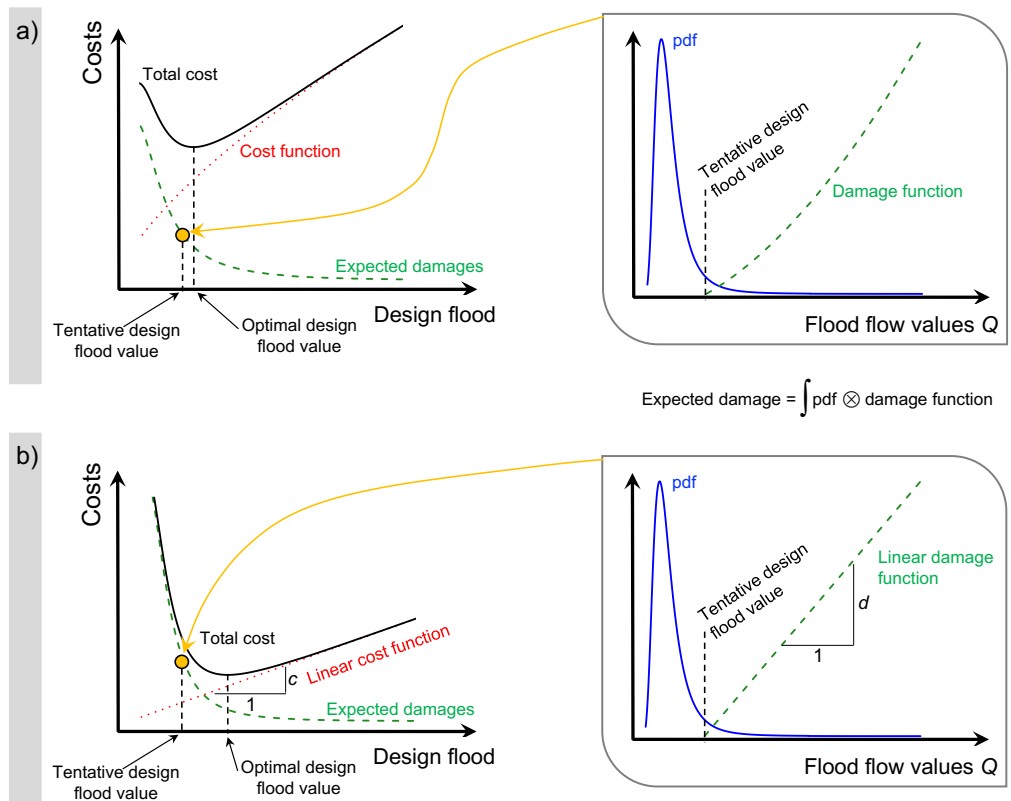

**Figure 1.** Illustrative example (without uncertainty) of the application of the cost-benefit framework to compute the design flood. Two generic cost and damage functions are reported in panel a, while panel b shows the linear functions adopted in the UNCODE framework.

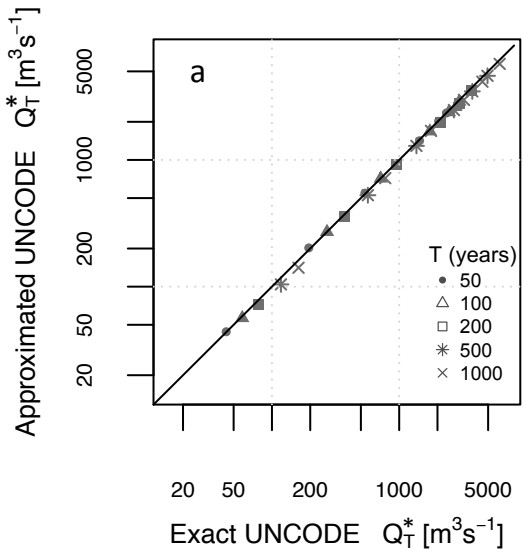

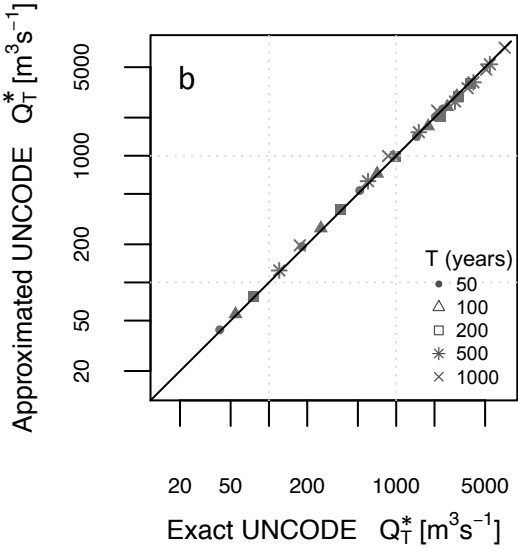

**Figure 2.** Comparison between the exact, $Q_T^*$, and the approximated, $\hat{Q}_T^*$, UNCODE estimators of the design flood for a pool of 6 flood records considered in Botto et al. (2014, Table 1) with at least 30 years of data. Different return periods are listed in the legend. The reference distribution used for this flood frequency analysis is the 3-parameter log-Normal (LN3) in panel "a" and the generalized extreme value (GEV) in panel "b".

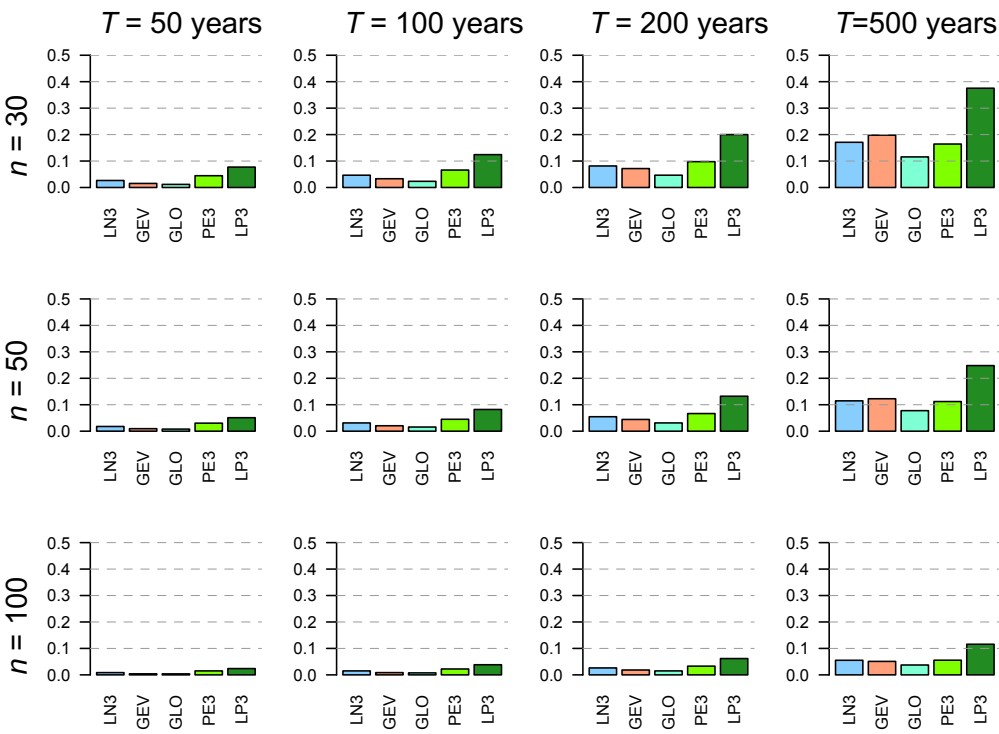

**Figure 3.** Values of the correction factor $\hat{y}$ from Eq. (6) for some values of the sample length $n$ and return period $T$ and for different three-parameter fitting distributions (LN3 = log-Normal; GEV= generalized extreme value; GLO = generalized logistic; PE3 = Pearson type III; LP3 = log Pearson type III). The LP3 corresponds to the PE3 with log-transformed variate.

**Table 1.** Coefficients to be used to estimate $\hat{y}$ based on the sample length $n$ and the return period $T$ (eq. 6) and corresponding regression diagnostics, for different 3-parameter fitting distributions (LN3 = log-Normal; GEV= generalized extreme value; GLO = generalized logistic; PE3 = Pearson type III; LP3 = log Pearson type III). The LP3 corresponds to the PE3 with log-transformed variate.

|  | $a_0$ | $a_1$ | $a_2$ | $R^2_{adj}$ | $MAE$ | $RMSE$ |
|---|---|---|---|---|---|---|
| LN3 | -0.82 | -0.25 | 0.809 | 0.94 | 0.0107 | 0.0160 |
| GEV | -2.27 | -0.3 | 1.110 | 0.85 | 0.0190 | 0.0321 |
| GLO | -2.36 | -0.25 | 0.994 | 0.85 | 0.0096 | 0.0145 |
| PE3 | 0.59 | -0.24 | 0.567 | 0.96 | 0.0080 | 0.0115 |
| LP3 | 0.78 | -0.26 | 0.687 | 0.89 | 0.0235 | 0.0363 |