# Peer review of "Technical Note: Design flood under hydrological uncertainty"

_Hydrology and Earth System Sciences, 2016_

## Referee Comment (RC1) · Anonymous Referee #1 · 5 Jan 2017

I find the topic and scope of the manuscript appropriate as a technical note in HESS. Simple yet technically sufficient approaches to include uncertainty in hydrologic design are of great significant in practice. However, the technical note is lacking critical details and requires clarification in some areas, which makes the manuscript difficult to fully assess at this time. The manuscript itself does not stand-alone and assumes that the reader is familiar with the details of the Botto et al. (2014) study. I would recommend major revision with additional review before publication in HESS. I have provided specific areas that need to be addressed with additional details or clarification.

More technical details are needed in the following areas:

- Parameters c and d are not well explained in the manuscript (p. 2, lines 17-18). The manuscript only indicates that they are site-specific and "influenced by topography

and land use among others...." A reader should have enough information in order to understand these parameters and how they are estimated without having to refer back to Botto et al. (2014).

- Eqn 15: C_TOT is not defined.

- p. 2, line 18: The authors state that setting C_TOT to zero gives the "optimal design flood value." Please help the reader connect why this is the case. Is this because this condition gives the local minima for C_TOT? Defining C_TOT will help with this point.

-p. 2, line 30: Please explain how c and d are related to the known value of the return period T

- The derivation of equation 5 and its relation to the pdf is not immediately obvious. Please provide a more detailed explanation of the origin of this equation and its conceptual meaning.

- The variable y needs to be better explained and defined as it serves as the focus of the contribution of the note, in my opinion. In the manuscript, y is only described as a non-negative number depending only on the pdf used to fit the flood frequency data (p. 3, line 12-13), which seems too simplistic. Please provide additional explanation. Later in line 29, the text states that y is estimated from regression, further confusing the reader as to how to interpret its meaning. The reader should not have to refer to the supplementary information to understand this.

- p. 3, line 20: It is unclear why the reference to Hosking and Wallis is placed at the end of this sentence. The reference placement implies that Hosking and Wallis (1997) have some comment as to the values of the a_j's, which is not the case. Remove the reference here but keep the reference in Table 1.

- More description is needed in the main text to discuss the computation of the a_j's and y values. Also, include in the main text the regressors used in the regression model.

Clarification is needed in the following areas:

- There needs to be a clarification in the introduction regarding the understanding of where uncertainty in the design flood arises. I agree that the design flood can be known as a single value when, as the authors state (p. 1, lines 18-19), "the frequency distribution and its parameters are known without uncertainty." My understanding is that the motivation for this work is because the frequency distribution and its parameters can only be estimated from a sample of flood data. It is for this reason, uncertainty arises and must be considered in practice. I recommend reworking lines 18-22 to something such as this:

"...period T. For practical reasons, Q_T is commonly expressed only as a single value; however, Q_T can only be expressed in this way if its frequency distribution and its parameters are known perfectly. In practice, one can only estimate the frequency distribution and its parameters using a sample of observed data, thereby creating uncertainty in the estimate of Q_T. However, the design of a hydraulic..."

- My understanding is that the authors only consider parameter uncertainty and not uncertainty in the choice of the pdf. Clarify this in the text.

Minor editorial comments:

p. 1, line 5: Avoid using a reference in the abstract.

p. 3, line 4: Change to read: "Botto et al. (2014) shows that the UNCODE..."

p. 5, line 2: Change to "increasing"
* * *

---

## Referee Comment (RC2) · A. Pugliese (Referee) · 12 Jan 2017

The paper "Technical Note: Design flood under hydrological uncertainty" by Botto et al., shows how to quickly find an estimate of hydrologic uncertainty to be added to classical statistical inference for the design flood. The paper is well written and is rather complete in all its sections, and it has the potential to be extremely useful for practitioners and engineers. I believe it is suitable for the publication in HESS as a technical note after some minor improvements, essentially due to miscommunicated reasonings, which, in my view, the authors might consider to take into account. Finally, I think there is enough material for another paper here, so I encourage the authors to consider deepening the analysis in the future, in order to find operational ranges and domains with real world data too, since the presented practical method needs to be as much robust as possible. For instance, an idea could be to extended the presented method using

regional flood frequency analysis, in order to overcome possible unsuitability of the presented procedure in data scarce regions.

**Minor comments**

1. The authors report that the parameters $c$ and $d$ are site-specific and controlled by topography and land use among others. These two parameters control the magnitude of the total cost function, and, ultimately, the design flood, but in most cases collecting cost data can be a cumbersome process, as they are usually unavailable. As far as I understood, the calibration of the empirical law (Eq. 5) does not need the knowledge of either $c$ or $d$, so that the resulting coefficients $a_0$, $a_1$, $a_2$, are rather general and independent from a specific site. Is it? Can the coefficients in Table 1 be used without any restriction on the location? I think the authors should address better the following thoughts:

   a) what is the role of the two parameters $c$ and $d$, how do they transfer their informations about site-specific cost rates to the parameters $a_0$, $a_1$, $a_2$, if any;

   b) or, is this transferring perhaps delivered by the parameters of the flood frequency curve only?;

   c) are the parameters $a_0$, $a_1$, $a_2$ general (independent by site, computed once and for all) or, perhaps, does the end-user have to fit the empirical law when needed on a specific site? If so, it would be useful to have a general point-by-point procedure, like an algorithm, to let the end-user implement it on a specific dataset or location;

   d I think the same sentence L20-23 P1 of supplementary material can be included into the manuscript, or at least the authors should mention exhaustively about the choice of such empirical expression for $y$.

2. The authors should consider to report the accuracy of the fitted empirical law in the body of the text too, at least for LN3 and GEV distributions.

3. In my view there is ambiguity in the mathematical notation between exact UN-CODE solution $Q_T^*$ and predicted (approximated) UNCODE, which is reported with the same variable $Q_T^*$. The authors might consider to change notation on one of the two, indeed the approximated UNCODE introduces one more source of error brought by the selected empirical law $y$.

4. L19 P3. The sentence in parentheses is put aside the main sentence, but I think it is rather important for the reader to know that regional analyses can be used where there is lack of data. The author should consider to expand the reasoning here, without parentheses.

**Notes and misspellings**

1. I agree with reviewer 1, I would remove the citation Botto et al., 2014 from the abstract to let it be more general.

2. L5 P3. Replace "methods" with "method".

3. L27 P3. I would add the range of variation of the index $j$, so "The coefficients $a_j$" will be "The coefficients $a_j$ with $j = 0, 1, 2$".

---

## Author Comment (AC1) · 16 Jan 2017

We acknowledge Alessio Pugliese and an anonymous reviewer for their valuable comments and suggestions. The major issue arising from these comments is that the description of the UNCODE procedure is not sufficiently detailed, thus making the paper not completely a stand-alone product.

Clearly, the core concepts of the UNCODE procedure are fundamental in the paper as the aim of the work is to provide a simplified (and easy-to-use) implementation of the method. Moreover, the "full" UNCODE procedure has been used in the manuscript to calibrate and validate its "simplified" implementation. We thus recognize that the reader must be able to fully understand the UNCODE procedure without relying on external material. On the other hand, the manuscript has been conceived to focus on

the simplified procedure and on its practical results, while giving only a summary of the UNCODE concepts. For this reason we have considered the technical note as a more adequate article type than a standard research paper.

In any case, these comments demonstrate that we were not fully successful in our attempt to synthetize the method: we are therefore going to prepare a revised version of the manuscript that will include a deeper revision of the description of the UNCODE procedure, by providing all the necessary elements to understand the model structure from the manuscript itself. Only the more technical results and the mathematical intermediate steps will be left to the reference of Botto et al. (2014) paper.

Last but not the least, the revised manuscript will implement a number of corrections and clarifications, following the suggestions by both reviewers.

---

## Author Comment (AC2) · 20 Mar 2017

[revised manuscript text omitted]

We thank the Referees and the Editor for their detailed analysis of the manuscript and useful comments. We report below the Referees' comments (italic font) along with our response (regular font).

The Referees will find the "Introduction" section of the manuscript deeply revised. While the reported concepts are the same as in the original manuscript, the new version is intended to give a concise, but more complete and stand-alone description of the UNCODE methodology, which was one of the major issues rised by both the Referees. Section 2 has been also reformulated to be clearer and to fix some misleading notations; moreover, details about the simulation study previously reported in the "Supplementary Material" are now included in the main text (the Supplementary Material is no longer provided with the current version of the manuscript being no longer necessary).

All the other general and specific issues are discussed below. Some questions, regarding the same topic, have been addressed together in the following response to provide a more comprehensive explanation.

*I find the topic and scope of the manuscript appropriate as a technical note in HESS. Simple yet technically sufficient approaches to include uncertainty in hydrologic design are of great significant in practice. However, the technical note is lacking critical details and requires clarification in some areas, which makes the manuscript difficult to fully assess at this time. The manuscript itself does not stand-alone and assumes that the reader is familiar with the details of the Botto et al. (2014) study. I would recommend major revision with additional review before publication in HESS. I have provided specific areas which need to be addressed with additional details or clarification.*

We thank the Referee for this comment and appreciate the opportunity to better clarify our research objectives and results. As mentioned above, the Introduction section has been completely revised to be stand alone, although references to Botto et al. (2014) are still used to refer to specific details. To better explain the UNCODE procedure, a graphical definition of the cost – benefit approach in its general form is provided by including a description in the text and a new figure (Fig. 1 in the revised manuscript). In particular, the new Fig. 1 specifies the cost and damage functions of a general cost-benefit framework (panel a) versus the functions used in the UNCODE context (panel b). This direct comparison is intended to better exemplify the hypotheses made in the model.

*More technical details are needed in the following areas:*

- *Eqn 15: C_TOT is not defined.*
- *p. 2, line 18: The authors state that setting C_TOT to zero gives the "optimal design flood value." Please help the reader connect why this is the case. Is this because this condition gives the local minima for C_TOT? Defining C_TOT will help with this point.*

The total cost function, C_TOT, is now defined and discussed in a more coherent way in the Introduction section. The computation of the optimal design value is also discussed in more detail. The new Fig. 1 helps the reader visualize the "optimal" value and its meaning (i.e., the minimum of the total cost function, which is the sum of costs and damages).

*- Parameters c and d are not well explained in the manuscript (p. 2, lines 17-18). The manuscript only indicates that they are site-specific and "influenced by topography and land use among others. . .." A reader should have enough information in order to understand these parameters and how they are estimated without having to refer back to Botto et al. (2014).*

*-p. 2, line 30: Please explain how c and d are related to the known value of the return period T*

These issues have been now explicitly discussed in the revised manuscript. We clarified that in general, in a cost-benefit framework, the cost and especially the damage functions are difficult to be estimated, the latter depending on a number of specific variables such as the exposure and vulnerability of the flooded goods, the topography, the land use, the flooding dynamics to mention but a few (page 2 lines from 10 to 14). This function turns out to be very site-specific and often unavailable for the lack of information needed to compute it. A new reference (Menoni et al, 2016) has been added to support this statement.

Moreover, the manuscript now reports that the UNCODE procedure makes use of simplified cost and damage functions. These have been reported in Fig.1b, and discussed throughout section 1. More details about the relationships between the classical flood frequency analysis and the cost – benefit approach are now provided, being a key concept in the UNCODE approach. A concise demonstration of the equivalence d/c=T is also provided.

*- The derivation of equation 5 and its relation to the pdf is not immediately obvious. Please provide a more detailed explanation of the origin of this equation and its conceptual meaning.*

*- The variable y needs to be better explained and defined as it serves as the focus of the contribution of the note, in my opinion. In the manuscript, y is only described as a non-negative number depending only on the pdf used to fit the flood frequency data (p. 3, line 12-13), which seems too simplistic. Please provide additional explanation. Later in line 29, the text states that y is estimated from regression, further confusing the reader as to how to interpret its meaning. The reader should not have to refer to the supplementary information to understand this.*

Section 2 has been also revised to facilitate the interpretation of the terms. Firstly, the notation has been changed to easily identify the full-UNCODE design flood (QT*) and the approximated-UNCODE design flood (hatQT*). According to this notation, also the correction factor "y" has an exact (y) and an approximated version (hat y).

The new notation, along with some more details moved from the Supplementary Material to the main text, helps to understand the origin of Eq. (5) (Eq. 6 in the revised version). Exact "y" values have been computed for a number of random samples and then a simplified equation (regression) has been used to compute its estimator (hat y) with minimal information: the sample length n (always know for at-site analysis, but equivalent n can be also evaluated at un-gauged sites) and the return period T.

*- More description is needed in the main text to discuss the computation of the a_j's and y values. Also, include in the main text the regressors used in the regression model.*

The relationship between y and the variables n and T has been obtained by means of linear regressions. Equation 5 shows the general formula to estimate y. There, the coefficients $a_0, a_1, a_2$ depend on the specific probability distribution function chosen for the inference procedure, which are reported in Table 1. Supposing we want to calculate the correction $y$ for the return period T and sample length $n$ with two different pdf: this can be pursued just choosing the appropriate coefficients from Table 1 and the same T and $n$ in both cases. Details of the simulation and regression procedure have been now fully reported in the main text. The Supplementary Material is thus no longer necessary.

*- p. 3, line 20: It is unclear why the reference to Hosking and Wallis is placed at the end of this sentence. The reference placement implies that Hosking and Wallis (1997) have some comment as to*

*the values of the a_j's, which is not the case. Remove the reference here but keep the reference in Table 1.*

We completely agree with the referee. The correction has been implemented.

*- Clarification is needed in the following areas:*

*- There needs to be a clarification in the introduction regarding the understanding of where uncertainty in the design flood arises. I agree that the design flood can be known as a single value when, as the authors state (p. 1, lines 18-19), "the frequency distribution and its parameters are known without uncertainty." My understanding is that the motivation for this work is because the frequency distribution and its parameters can only be estimated from a sample of flood data. It is for this reason, uncertainty arises and must be considered in practice. I recommend reworking lines 18-22 to something such as this:*

*". . .period T. For practical reasons, $Q\_T$ is commonly expressed only as a single value; however, $Q\_T$ can only be expressed in this way if its frequency distribution and its parameters are known perfectly. In practice, one can only estimate the frequency distribution and its parameters using a sample of observed data, thereby creating uncertainty in the estimate of $Q\_T$. However, the design of a hydraulic. . ."*

We thank the referee for the suggestion which makes the point clearer; we included the correction in the Introduction section of the manuscript.

- *My understanding is that the authors only consider parameter uncertainty and not uncertainty in the choice of the pdf. Clarify this in the text.*

It is correct. The point, originally discussed in Sect. 3, has been clarified in page 5 lines 17-20; the procedure does not include the issue of uncertainty in model selection, but provides a way to compute the UNCODE estimator tailored for different distributions commonly used for this kind of analysis.

*Minor editorial comments:*

*p. 1, line 5: Avoid using a reference in the abstract.*

Amended

*p. 3, line 4: Change to read: "Botto et al. (2014) shows that the UNCODE. . ."*

Amended

*p. 5, line 2: Change to "increasing"*

Amended
*The paper "Technical Note: Design flood under hydrological uncertainty" by Botto et al., shows how to quickly find an estimate of hydrologic uncertainty to be added to classical statistical inference for the design flood. The paper is well written and is rather complete in all its sections, and it has the potential to be extremely useful for practitioners and engineers. I believe it is suitable for the publication in HESS as a technical note after some minor improvements, essentially due to miscommunicated reasonings, which, in my view, the authors might consider to take into account.*

*Finally, I think there is enough material for another paper here, so I encourage the authors to consider deepening the analysis in the future, in order to find operational ranges and domains with real world data too, since the presented practical method needs to be as much robust as possible. For instance, an idea could be to extended the presented method using regional flood frequency analysis, in order to overcome possible unsuitability of the presented procedure in data scarce regions.*

We would like to thank the reviewer for the useful comments to improve the paper and for the kind encouragement. We have addressed all the comments as explained below.

*Minor comments*

*The authors report that the parameters c and d are site-specific and controlled by topography and land use among others. These two parameters control the magnitude of the total cost function, and, ultimately, the design flood, but in most cases collecting cost data can be a cumbersome process, as they are usually unavailable. As far as I understood, the calibration of the empirical law (Eq. 5) does not need the knowledge of either c or d, so that the resulting coefficients $a_0$, $a_1$, $a_2$, are rather general and independent from a specific site. Is it? Can the coefficients in Table 1 be used without any restriction on the location? I think the authors should address better the following thoughts:*

The Referee has brought up some good points and we appreciate the opportunity to clarify better our research objectives and results.

- *what is the role of the two parameters c and d, how do they transfer their information about site-specific cost rates to the parameters $a_0$, $a_1$, $a_2$, if any;*

A detailed discussion of the role of the parameters c and d has been included in the revised version of the manuscript (Section 1) to make the theoretical description of the UNCODE procedure concise, but

completely stand-alone. More details about the meaning and the estimation of a0, a1 and a2 are now included in section 2.

In general, in a cost-benefit framework the cost and, especially, the damage functions are difficult to be estimated, the latter depending on a number of specific variables such as the exposure and vulnerability of the flooded goods, the topography, the land use, the flooding dynamics to mention a few (page 2 lines from 10 to 14). This function turns out to be very site-specific and often unavailable for the lack of information needed to compute it. A new reference (Menoni et al, 2016) has been added to support this statement. This point has been clarified in the revised manuscript including a new figure (Fig. 1) that sketches the cost-benefit procedure and help visualizing the involved functions. The main text has been integrated accordingly.

With respect to a general cost-benefit analysis, the UNCODE procedure adopts specific cost and damage functions (linear with slope c and d) that are also included in the new Fig.1 for an easy comparison with the general cost and damage functions. The revised version of the Introduction section also reports a more detailed description of the derivation of the equivalence d/c=T that, in practice, makes the estimation of c and d values unnecessary. This point has been reported on page 3 lines 10-14 …, highlighting that these parameters influence only the magnitude of the integral in Eq. (3) (i.e., the total expected costs under uncertainty), while its minimum (i.e., the UNCODE design flood) is unaffected.

Moreover, it is true that both sets of parameters $(c, d)$ and $(a_0, a_1, a_2)$ involve T; however, the two sets of parameters derive from different concepts. In the first case the use of the return period is a consequence of the application of the cost-benefit framework under some simplifying assumption and it can be considered as an element of the UNCODE model (as already derived by Botto et al., 2014). In the second case, the return period is a proxy (together with n) of the quantile uncertainty (i.e., it considers the typical spread of the confidence bands of the flood frequency curve with the return period).

- *or, is this transferring perhaps delivered by the parameters of the flood frequency curve only?*

- *are the parameters $a_0$, $a_1$, $a_2$ general (independent by site, computed once and for all) or, perhaps, does the end-user have to fit the empirical law when needed on a specific site? If so, it would be useful to have a general point- by-point procedure, like an algorithm, to let the end-user implement it on a specific dataset or location;*

The estimation of the UNCODE design flood is a non-linear process that depends on the sample uncertainty and how it propagates to the distribution quantiles. The proposed method aims at simplifying this complex process by requiring only the estimate of the standard quantile QT and the multiplying factor y that includes all the elements that drive the "propagation" of the uncertainty to the distribution quantile. The developed approach is thus general so that the regression parameters $(a_0, a_1, a_2)$ are computed once and for all, although they can be considered accurate only within a subset of the T-n-Lmoments space, as specified in the main text of the revised manuscript.

The application still depends on the choice of the distribution as the UNCODE method does not account for the model uncertainty. However, this does not limit the application of the procedure as different distributions can be tested as in the classic flood frequency analysis.

*I think the same sentence L20-23 P1 of supplementary material can be included into the manuscript, or at least the authors should mention exhaustively about the choice of such empirical expression for y.*

The description of the regression has been detailed in the revised manuscript (sec. 2), including the information originally reported in the Supplementary Material (the Supplementary Material is no longer provided with the current version of the manuscript).

*The authors should consider to report the accuracy of the fitted empirical law in the body of the text too, at least for LN3 and GEV distributions.*

This suggestion has been implemented in the revised manuscript, reporting a more complete analysis of fitting residuals.

*In my view there is ambiguity in the mathematical notation between exact UNCODE solution QT and predicted (approximated) UNCODE, which is reported with the same variable QT . The authors might consider to change notation on one of the two, indeed the approximated UNCODE introduces one more source of error brought by the selected empirical law y.*

The notation has been revised to properly identify the exact and the approximated values of both $\hat{y}$ and $\widehat{Q_T^*}$. This is now highlighted in the first part of section 2.

*L19 P3. The sentence in parentheses is put aside the main sentence, but I think it is rather important for the reader to know that regional analyses can be used where there is lack of data. The author should consider to expand the reasoning here, without parentheses.*

We thank the Referee for the note, the sentence has been modified.

*Notes and misspellings*

*I agree with reviewer 1, I would remove the citation Botto et al., 2014 from the abstract to let it be more general.*

Amended

*L5 P3. Replace "methods" with "method".*

Amended

*L27 P3. I would add the range of variation of the index j, so "The coefficients aj"  will be "The coefficients aj with j = 0, 1, 2".*

Amended, index j has been removed and coefficients are now explicitly defined, $a_0, a_1, a_2$

[revised manuscript text omitted]

---

## Author Response (AR3)

*We thank the Referee and the Editor for their detailed analysis of the manuscript and useful comments. We report below out point-to-point response (italic font) to the Referee's comments (regular font).*

The authors have done a good job incorporating the review comments into the manuscript and I feel the content of the manuscript is now acceptable for publication. I do, however, have a number of editorial comments that I hope will help clarify the manuscript further. Therefore, I am recommending acceptance of this manuscript with minor revisions.

p. 1, line 3: The statement is made: "the design flood value is no longer a deterministic value, but should be treated *as a random variable itself*." I disagree with this statement. The design flood is treated as a random variable in its estimation; the point is that this component is often ignored when one value is reported for the design flood. This should be revised. Once potential revision could be (in place of lines 2-5): "However, when hydrologic uncertainty is accounted for, the design flood is defined **as a range of values."**

*The sentence has been revised to incorporate the suggestion and now reads: "However, when hydrologic uncertainty is accounted for, the design flood value for a specific return period is no longer a unique value, but is defined as a distribution of values." We prefer to use "distribution" instead of "range" because the method actually operates on the distribution of the quantile estimate.*

Abstract, Paragraph 2: The wording is awkward here and needs to be fixed. It is not that the design flood is "ambiguous" but that, with uncertainty considered, the design flood could be a range of values. The UNCODE procedure uses a numerical cost-benefit approach to capture uncertainty but still result in a single design flood value. The advance of this paper is a simple correction factor that can replace the numerical computations in UNCODE. Paragraph 2 should be revised to reflect these points.

*Both points have been highlighted in the revised version. In particular, the "ambiguity" is now explained as: "The Uncertainty Compliant Design Flood Estimation (UNCODE) procedure is a novel approach that, starting from a range of possible design flood estimates obtained in uncertain conditions, converges to a single design value." To stress that the numerical burden has been replaced with a simple procedure, we now report: "This is obtained through a cost-benefit criterion with additional constraints that is numerically solved in a simulation framework. This paper contributes to disseminate the UNCODE procedure without resorting to numerical computation, but using a correction coefficient that modifies the standard (i.e., uncertainty-free) design value on the basis of sample length and return period only."*

p. 1, line 12: Change "This new design tool" to "UNCODE, when coupled with this simple correction factor provides…"

*Amended*

p. 2, line 13: Replace "for" with "due to"

*Amended*

p. 2, line 15: Add a phase to the end of the statement "the cost-benefit method thus appears as an attractive design approach" since you do not yet explain why this would be the case. Also, for the second part of the sentence, revise to read: "however, due to difficulties in determining the cost and damage functions, it is often unable to be applied." This sentence is confusing overall and disjointed from the previous paragraph.

*We agree that the sentence does not add any further information and can be to some extent confusing, so both sentences have been deleted. However, the concept that "… the cost-benefit method often results inapplicable." has been moved to the previous paragraph to complete the discussion on the cost-benefit method.*

Equation 1 and descriptions of c and d: The parameters c and d are still not defined sufficiently. It seems from Equation 1 that they do not change; yet if they are piecewise linear functions, one would expect that they would have subscripts to define the portions of the lines that each slope applies to. Clarify this. Also, add an explicit definition of c and d to the descriptions of equation variables directly after Equation 1.

*The meaning of the coefficients c and d is recalled by the reference to Figure 1b just before Eq. (1) that shows the shape of the (simplified) cost and damage function. The word "piecewise" has been deleted to avoid confusion: actually, the cost function is strictly linear and also the damage function is strictly linear within the limits of integration Q\*-Infinity. More details on the mutual dependence of the two coefficients are left to the following paragraph that illustrates the d/c=T equivalence.*

p. 2, line 32: Delete "these findings through"

*Amended*

p. 3, line 2: The authors note that the framework can be extended "due to the limited data availability" yet there is nothing more mentioned about this. I would delete this statement or expand here. Otherwise, it seems out of place and it is unclear what the authors mean.

*We thank the reviewer for this comment. The sentences have been slightly changed, but we believe that the meaning is now much cleared. First, the revised version specifies that the "sample uncertainty" is what the UNCODE considers, without referring to "data availability"; second, we clarify that the cost-benefit is easily applicable to uncertain conditions, thus also the UNCODE (being a special cost-benefit method) can easily account for the same kind of uncertainty.*

p. 3, line 15: Change to read "…approximated and reliable method…"

*Amended*

p. 3, line 19: Revise to read: "…results in a systematically larger value…" and then add a phrase explaining why this is the case.

*Correction has been implemented. Regarding the explanation of this effect, the reference to Botto et al. (2014) has been anticipated (originally it was reported only after eq. 4) in order to avoid to report the discussion of the previous paper and to keep the paragraph focused on the aim of the present work.*

p. 4, line 3: Coefficients a0, a1, and a2 are still not clearly defined. Are these the parameters of each of the distributions? I think this is explained in the review response but not in the text.

*This point has been specified in the revised manuscript.*

p. 4, line 5: Instead of call the distributions a "list," which sounds arbitrary, state that you considered the most common distributions used in flood frequency.

*Amended*

p. 4, line 8: Change "variates" to "values"

*Amended*

p. 4, line 10: Be more precise in your wording. Revise to: "Looking at the properties of the moments of the distribution in L-moment space,…"

*Amended*

p. 4, line 14: Change "fitting" to "fitted" and consider instead of F to signify the sample distribution, to use P_hat, which is the estimate of the parent distribution.

*We prefer to use F to keep a clear distinction between the distributions used to create the reference dataset for analysis (P) and the distribution which can be adopted by the analyst to perform the frequency analysis (F)*

p. 4, line 30: Change to read "distribution results in the best performance in terms of …" and explain why this is the case for PE3.

*Amended. Reference to numerical values in Table 1 is now reported.*

p. 5, line 13-16: Same comments in general as the abstract. These paragraph needs to be revised. One possible revision to consider: "…analysis provides a solution to the estimate of the design flood when considering the uncertainty in the quantile estimation; however, application…and computationally demanding for the practitioner. An approximate but reliable framework….to easily compute the UNCODE design flood value from the standard value using a correction factor, y_hat."

*The suggestion has been implemented in the text.*

p. 5, line 18: Change "standard" to "traditionally-computed" and add "…distributions in flood frequency analysis, they… "

*Amended*

p. 5, line 19: Change "management" to "quantification"

*Amended*

p. 5, line 20: Change "preliminary" to "methods of"

*Amended*

p. 5, line 22-23: Remove the sentence starting with "This implies that…" The point made here is too confusing to the reader and unclear.

*We prefer to keep the concept in the text as it serves to introduce the idea of the "value of data". However, the sentence has been revised to make it clearer.*

p. 5, line 30-31: The statement is made that the values lead to a "reduced UNCODE design flood." I would caution the authors to make such a general statement that could leader the reader to believe that reduced flood design values have some type of benefit overall. What do the authors intend here in this sentence. Please add text to clarify.

*The text has been re-organized by moving the concept of "value of data" to the previous sentence "The obtained results demonstrate that an increase…", which has also been modified to clarify that the "reduction" is intended between the UNCODE computed with few data and the UNCODE computed with a larger dataset (and not a reduction in the standard design flood)*

p. 6, line 4: After semicolon, add "therefore"

*Amended*

Supplementary material: Same comment as above when referencing the "list" of distributions considered.

*Supplementary material is no longer provided as the relevant information has been included in the main text.*

[revised manuscript text omitted]